# Carbon Emissions during the Building Construction Phase: A Comprehensive Case Study of Construction Sites in Denmark

Kai Kanafani *, Jonathan Magnes, Søren Munch Lindhard and Maria Balouktsi

Department of the Built Environment, Aalborg University, 2450 Copenhagen, Denmark;
sml@build.aau.dk (S.M.L.); mariab@build.aau.dk (M.B.)
* Correspondence: kak@build.aau.dk

**Abstract:** Buildings are major contributors of carbon emissions and related global warming. Emissions occur along all building stages, from a whole-life perspective, including material production, construction processes, building operations, maintenance and end-of-life processes. Upfront emissions include processes before building operations. They can be influenced immediately and will have a positive effect today. However, mitigation potentials during the construction stage are often overseen in research. This study presents an analysis of the carbon emissions of 61 Danish construction sites based on their energy consumption, waste production (module A5) and transport to site (A4). The results show carbon emissions for A4 of 0.28 and for A5 of 1.00 $kgCO_2e/m^2$ gross floor area per year over 50 years. This is 13.47% of the Danish whole-life carbon reference of 9.50 $kgCO_2e/m^2y$, which includes the product stage (A1–3), replacements (B4), operational energy use (B6) and waste processes and disposal (C3–4). Almost half of the emissions are related to construction waste followed by electricity, heat and fuel. Floor area and building use have not shown to be influential for carbon emissions, suggesting other parameters are more important. The significance of modules A4 and A5 suggests implementing them in future whole-life carbon assessments and related policies. This paper also demonstrates the development of generic emission coefficients, which are suited to increase the feasibility for application in the building industry. Finally, the usability of module A4 and A5 in environmental product declarations is discussed.

**Keywords:** construction process; transport; LCA; whole-life carbon assessment; upfront emissions

## 1. Introduction

### 1.1. Background

Buildings are major contributors of carbon emissions and related global warming. Having adopted the Paris agreement for limiting global warming to 2.0 degrees, many countries are taking measures for reducing carbon emissions in all sectors. The first countries have now launched national whole life cycle carbon regulations for buildings [1]. A building's life cycle includes the production of materials, the construction process, building operations and maintenance, and its end-of-life processes. Mitigating emissions released before the building operation begins, commonly referred to as upfront carbon emissions [2], has an immediate positive impact, since time is running out for meeting the global warming targets, as the latest report by the Intergovernmental Panel on Climate Change (IPCC) warns [3]. Although the construction process stage belongs to upfront emissions, which can be mitigated today, this building life cycle stage is often overseen in research.

### 1.2. Life Cycle Assessment and Construction Processes

This study analyses carbon emissions associated with processes including energy consumption, construction waste and transport in life cycle modules A4 and A5, see Figure 1. Whole-life carbon assessments of buildings are receiving increasing societal attention in countries with strict environmental regulation. They are based on the life

cycle assessment (LCA) method defined in technical standards such as the EN 15978 [4] for buildings and EN 15804 [5] for building products. With the narrow focus on greenhouse gas (GHG) emissions, named carbon emissions in this paper, this relatively complex assessment method has become accessible for voluntary certification schemes and recently national policy. The reason for the increasing adoption is both the unambiguous 2.0-degree Paris target [6] and the now-available harmonized methods, environmental product data and calculation tools.

| Upfront emissions (now) | | | | | Future emissions (scenarios) | | | | | | | | | | |
|---|---|---|---|---|---|---|---|---|---|---|---|---|---|---|---|
| Product stage | | | Construction proces stage | | Use stage | | | | | | | End of life stage | | | |
| Raw material supply | Transport | Manufacturing | Transport | Construction installation proces | Use | Maintenance | Repair | Replacement | Refurbishment | Operational energy use | Operational water use | De-construction demolition | Transport | Waste processing | Disposal |
| A1 | A2 | A3 | A4 | A5 | B1 | B2 | B3 | B4 | B5 | B6 | B7 | C1 | C2 | C3 | C4 |

**Figure 1.** Transport of construction products to building sites and on-site installation processes (see black box) belong to upfront processes responsible for emissions before building operations start and can be mitigated in the short term. Building life cycle stages and modules after EN 15978 [4].

Broadly, a building's life cycle is divided into an embodied and an operational part. The latter involves the carbon emissions associated with the operational energy use (B6) and the operational water use (B7). All other life cycle modules belong to a building's embodied carbon footprint. The standard also views the construction process stage, which includes transport to site in module A4 and the installation process in module A5, as scenarios. This classification is problematic, because it underrates the possibility of carbon mitigations through societal awareness and policy making. Therefore, we propose to extend the upfront boundary encompassing all emissions associated with processes occurring before building handover (Figure 1). This is according to the updated EN 15643:2021 [7] and is also expected to be implemented in the ongoing EN 15978 revision.

Recent meta studies demonstrate that embodied carbon emissions account for 64% [8] or 20–90% (depending on the energy efficiency) [9] of the whole building life cycle [10]. Despite ongoing innovations in material technology, such as the increased use of bio-based materials and design optimization, operational impacts are currently decreasing more quickly due to the energy sector's gradual decarbonisation. This development increases the significance of upfront impacts even further in the near future. Along with their relative importance, another crucial aspect to consider is that upfront emissions are immediately reducing the remaining global carbon budget and are 'locked-in' during construction without any possibility to influence them after completion. For these reasons, a greater priority to upfront emissions is expected to be given in science and policy development.

The reason for including A4 and A5 in LCA is to improve the accuracy of assessments for achieving a better steering effect towards mitigating emissions. Leaving modules A4 and A5 out of the equation will omit some of the upfront environmental impacts, which

actually can be reduced through the optimization of building design and execution. A more complete assessment also helps avoiding burden shifting, where impacts are moved from included modules to those which are not included.

A key example is the differentiation between prefabrication and in situ building. By including A5, the efficiency of prefabrication due to shorter installation times, less wastage as well as less intermediate heating and drying, will be reflected in the results. Also, the difference between prefabricated elements and individual products affects transportation in module A4. Flat concrete elements often have a low-capacity utilization of trucks compared with timber elements or individual products. On the other hand, elements are mostly transported directly to site, while other products may include multiple deliveries on a circular route. This, however, does not always mean low impacts from transport, because prefabrication often requires extra transport of individual products from the manufacturer to the prefabrication factory, which adds to the subsequent transport of the element to the construction site.

Including A4 and A5 is therefore important for improving assessments and avoiding unwanted side-effects, which may compromise the environmental targets. A certain optimization potential at the project level is assumed since the geographic origin of supplies and the installation process can be influenced. Changes at the system level such as energy supply, transport efficiency or construction method may add to the expected mitigation potential.

*1.3. Existing Standards and Limit Values*

Several European countries have now set binding requirements to report whole-life carbon assessment results of buildings, and some have even introduced or will soon introduce binding embodied carbon or whole life cycle limits. These countries are Denmark, Finland, France, The Netherlands, Norway and Sweden, and while in some countries, national regulations are not yet in force, local authorities with regulatory powers take the lead and demand whole-life carbon assessments, such as the Greater London Authority.

Despite differing with respect to the required assessment scope, all mandatory requirements either already include A4 and A5 in their scope or currently investigate ways of, and implications for, including these two modules in future requirements. For example, this is the case for the Danish building regulations currently being limited to A1–3, B4, B6 and C3–4.

As the consideration of the A4 and A5 modules in whole-life calculations within a more formal context is recent, some methods provide related default values to assist users that lack more product-specific information, especially in the early design steps. These default values are usually derived from studies of real construction sites. For example, the official database for generic emission data in Finland provides a value of 27 $kgCO_2e/m^2$ of the building floor area for A4, whereas default values for A5 range from 46 to 78 $kgCO_2e/m^2$, depending on the building type (i.e., residential, office, school and kindergarten) and excluding earthwork; for the latter, a reference value of 7 $kgCO_2e/m^2$ is given [11].

Another example is the RICS methodology, which is the base method for the binding assessment requirements established by the Greater London Authority. The new draft of the RICS (currently under consultation) divides A5 into four sub-modules that represent different types of activities—pre-construction demolition, construction activities, waste and waste management and worker transport. For the first three types of activity, the RICS provides reference values to be used until site-specific data are available: 50, 25 and 5 $kgCO_2e/m^2$, respectively [12].

It becomes clear that, in the interest of simplification while maintaining completeness, the provision of average and default values for A4 and A5 is an essential approach to compensate for the lack of product- and site-specific information.

*1.4. Existing Studies on Carbon Emissions in Modules A4 and A5*

In Denmark, the construction industry is responsible for the generation of 12 million tons of waste, corresponding to 58% of the total waste generation [13]. When looking at carbon emissions, the construction industry is responsible for 30% [14] of all Danish

GHG emissions. Therefore, the exploration of direct and indirect emissions associated with building construction through the lens of a whole-life perspective has also gained an increased research focus.

Most research on whole-life carbon emissions of buildings extend the conventional focus on operational energy with emissions embodied in building materials [15–17]. Only few studies investigate emissions generated from transporting materials to sites (module A4) or from construction site processes (module A5). Table 1 provides an overview of existing quantitative results for these modules.

**Table 1.** Existing studies with upfront GHG emissions, which specifically indicate results in modules A4 and A5.

| Reference (by Year) | Characteristics of Building Case(s) | | | | | | Upfront Emissions (kgCO$_2$e/m$^2$y) | | |
|---|---|---|---|---|---|---|---|---|---|
| | Country | Building Type | Floor Area (m$^2$) | Storeys above Ground | Basement Included | Included Stages | A1–3 | A4 | A5 |
| Yan et al., 2010 [18] | Hong Kong | Office | 42,000 | 30 | - | A1–5 | Only percentages | | |
| Monahan & Powel 2011 [19] | UK | Semi-detached house | 91 | 2 | No | A1–5 | 6.51 | 0.16 | 1.43 |
| Takano et al., 2014 [20] [1] | Germany Finland Italy | Multi-family residential | 726 730 1840 | 5 3 5 | Yes No No | A1–5, B6 | Only percentages | | |
| Seo et al., 2016 [21] | Korea | Mixed use | 5556 | 4 | Yes | A1–5 | 15.19 | 0.39 | 0.67 |
| Padilla-Rivera et al., 2018 [22] | Canada | Multi-family residential | 1512 | 4 | No | A1–5 | 4.12 | 0.72 | 0.66 |
| Petrovic et al., 2019 [23] | Sweden | Detached house | 180 m$^2$ incl. 30 m$^2$ garage | 2 | No | A1–5, B1–6, C1–4 | 3.40 | 0.05 | 0.45 |
| Karlsson et al., (2021) [24] | Sweden | Multi-family residential (4 variants) | 2198 | 6 | No | A1–5 | 3.97 average | 0.87 average | |
| Sezer & Frederiksson 2021 [25] | Sweden | Varies (40 cases) | Varies | Varies | Varies | A1–5 | - | 0.33 | - |
| Röck et al., (2022) [10] [2] | Belgium, Denmark, Finland, France, The Netherlands | Varies (769 cases) | Varies | Varies | Varies | A1–5, B1–4, C1–4 | 6 average | 0.8 average | |
| This study | Denmark | Varies | Varies | Varies | Varies | A4–5 | - | 0.28 1.28 | 1.00 |

[1] Results estimated from source figure; [2] Meta study.

Yan et al. [18] studied the carbon emissions of construction by following a 42,000 m$^2$ 30-storey office and retail building project in Hong Kong. The project used reinforced concrete as its load-bearing structure. The case study showed that 82–86% of all emissions stemmed from the construction materials, 6–8% stemmed from transportation while 6–9% stemmed from the on-site construction processes.

Monahan and Powel [19] followed a 91 m$^2$ low-energy housing project in the UK. The house was constructed as a two-floor building construction using module-based timber structures. The study found the following contributions to carbon emissions: construction materials—79.92%, transportation—2.41%, waste—13.47%, heating—0.62%, electricity—2.59% and diesel—0.99%.

Takano et al. [20] employed three multi-storey wooden residential buildings in different locations in Europe to investigate the GHG emissions associated with their construction process relative to the other life cycle stages: the results showed that the construction

stage accounts for 20–30% of the upfront emissions, with the share of A4 ranging from approximately 30% to more than 50% of A4–5 emissions.

Seo et al. [21] followed a Korean office and apartment project at approximately 2000 m². The project included one under-ground floor and four over-ground floors, and the building structure was a mixed steel and reinforced concrete structure. In this case, the carbon emissions from the material production stage constituted 93.4% of the upfront emissions, while on-site construction constituted 4.2% and transportation constituted 2.4% of those emissions. In a Canadian study, Padilla-Rivera et al. [22] studied a 1500 m² four-storey residential building with 20 apartments in a prefabricated timber structure. The study revealed that 75% of the emissions were related to the production materials, transportation constituted 13%, waste constituted 11% while the on-site works constituted 1%.

Regarding studies from the Nordic region, Petrovic et al. [23] analysed a wooden single-family demonstration house of 180.4 m² (house, 150.4 m² and garage, 30 m²) located in Sweden, distributed over a ground floor and upper floor. Their findings suggested that the construction process stage represents 13% of the upfront emissions. In a more recent study from Sweden, Karlsson et al. [24] used the LCA method performed by Erlandsson et al. [26] in which five different construction designs were applied to a reference building (a five-floor residential building in Stockholm, amounting to 2198 m²). The designs included prefabricated concrete and wooden system variants. The study found that, on average, the construction process stage (A4–5) accounts for 18% of the upfront emissions, with material transports making up a larger share of the emissions for the prefabricated systems.

As the authors of [25] indicate, the previously mentioned works are based on single case studies and all studies have focused on identifying emissions at a detailed level. One of the few limited studies that include ranges and mean values for the construction process stage based on a great number of diverse building cases from different countries reports a mean contribution of slightly above 10% to the upfront emissions [27]. Overall, there is a huge deviation in the findings, where the emission caused by transportation and on-site works differ by 6.6–25%.

This said, the cases were very different in nature and were located in very different places, thus culture and processes might have a huge effect. Moreover, there might be differences in how and what is measured when estimating carbon emissions. En et al. [28] lists the various data-extraction practices and assumptions made in studies quantifying carbon emission of A4 and A5 modules (among other modules) based on a review of 65 articles. For example, Soust-Verdaguer et al. [29] found that the LCA result for the A4 module of an Austrian case study can vary by approximately 30% when employing different modelling options. A similar order of variation in A4–5 value is seen in Fufa et al. [30] when comparing the as-design value with the as-built one of a campus building in Norway. The latter is more than 25% larger. Nevertheless, the above clearly shows the need for a large case study involving multiple cases to increase the knowledge on emissions related to the A4 and A5 modules and to make comparisons across project types.

Another Swedish study with 40 cases [25] analyses the number of transports per floor area and the potential reduction through optimized logistics. They found a huge variation between 0.04 and 1.33 transports/m² floor area. Combining the actual number of transports with average emissions, they estimate a transport share of at least 10% of GHG emissions in Swedish housing projects.

## 2. Materials and Methods

### 2.1. Overall Approach and Case Selection

The present study follows a quantitative approach with a correlational research design aimed at identifying relationships between multiple and various factors [31]. The research design shall establish a reliable relationship between carbon emissions during the construction stage, building type and floor area. Moreover, by quantifying the carbon emissions associated with the construction process stage, the aim is to identify the significance of the A4 and A5 modules compared to the remaining life cycle carbon emissions in construction.

All cases are recently completed new buildings varying in scale, use and site conditions. The cases were selected primarily based on the availability of data in modules A4 and 5, which resulted in two separate sets of cases.

As noted in [32], the system boundary (Figure 2) follows the EN 15978 structure but only includes activities with major impacts. See a description of the boundary and the applied scenarios in the following sections for A4 and A5.

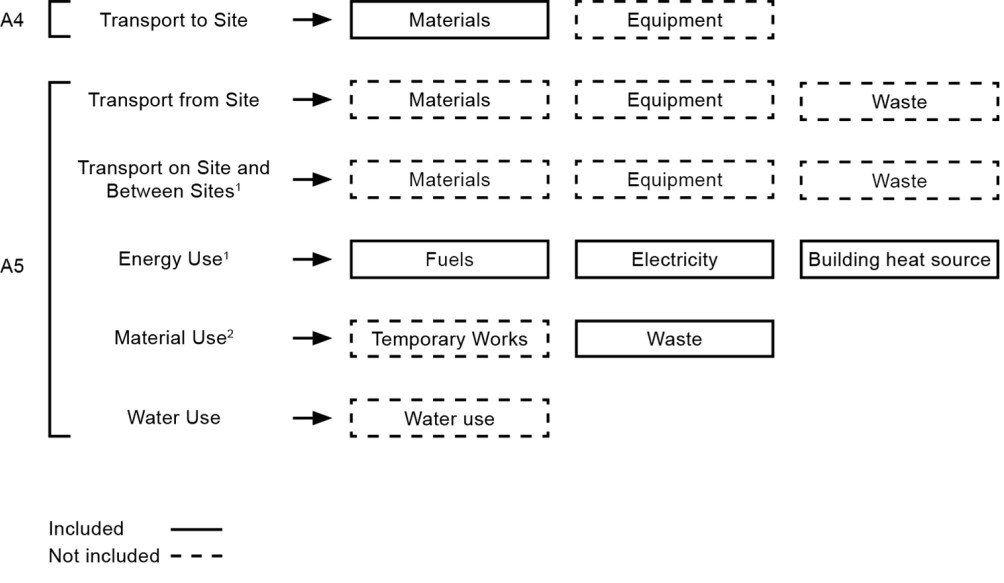

**Figure 2.** Applied system boundary.

## 2.2. Method for Module A4

Specific transport distances of the actual purchased products in building projects are not available in existing project data or statistics. Despite that invoices and delivery notes exist for all materials and equipment delivered to the site, they do not provide accurate and sufficient data for assessing transport. More specifically, they do not include possible previous transports from the manufacturer to storage or the whole seller. Furthermore, the given addresses do not always represent the outgoing location but the supplier's sales department. A manual study on mapping the complete transport chains for all products in specific cases would therefore not be possible on the basis of existing data and would have required a large number of interviews or the collection of secondary data on the actual routes.

For achieving a consistent method, the following semi-generic approach was chosen. Initially, building products were classified into 17 groups and 36 subgroups (Table 2) based on the hypothesis that emission coefficients must be differentiated into groups of potentially varying origins on the Danish market. In a subsequent step, transport information for all subgroups were collected from available data sources, including declared A4 modules in representative environmental product declarations (EPDs) and statistics collected from companies and industry associations.

Based on multiple data sources reflecting a representative market share, a mean value was derived for the transport distances and Global Warming Potential (GWP), defined for per kg product in each subgroup. This showed that an increasing number of EPDs include A4, however some do not provide specific results but provide an impact for a generic distance such as 100 km, which then has to be scaled to specific project conditions. While this is useful for achieving more accurate assessment in specific building LCA, this data could not be used for our purposes. In the case of unspecific EPDs data, module A4 was calculated based on the manufacturer location and the estimated transport to a Danish building site. These assumptions were based on actual industry specifications to the degree

possible. Subgroups including a share of imports through road transports were estimated by calculating the distance from the factory to Odense, a destination in the geographical centre of Denmark.

**Table 2.** Classification of product groups and subgroups for assessing transport in module A4.

| Group | Subgroup |
| --- | --- |
| Concrete | Ready-mix |
| | Wall/floor slab elements |
| | Other precast elements |
| Timber | Bars |
| | Boards (particle, OSB, plywood), planks, flooring |
| | Elements |
| Steel | Reinforcement bars, nets, prestress wires |
| | Sheets and profiles |
| Aluminium | Sheets and profiles |
| Gypsum | Boards |
| Tiles and bricks | Brick |
| Tile stone | Roof tiles |
| Cementitious products | Aerated concrete blocks |
| | Lightweight concrete blocks |
| | Fibre cement boards |
| | Cementitious mortar and render |
| Calcium-silicate | Sand-limestone |
| Zinc | Zinc sheets |
| Bituminous products | Roofing felt |
| Openings | Windows and doors |
| | Curtain wall facades |
| Stone | Natural stone |
| Insulation | EPS |
| | Calcium-silicate |
| | Cellulose |
| | Wood fibre |
| | Mineral wool |
| Membranes and coatings | Vapor barrier |
| | Paint |
| Building services | Photovoltaic panels |
| | Ventilation components |
| | Heating components |
| | Mechanical components |
| | Water and sewage system components |

For overseas imports, trans-shipment was assumed to take place at the port of Hamburg unless otherwise stated. Finally, transport for technical building services was based on an estimated distance of 500 km due to a lack of aggregated data in this complex product group.

These derived emission coefficients for product subgroups were then used for calculating module A4 in nine construction cases, where they were combined with specific building product quantities. The case number is viewed sufficient for generating results that provide an indication of the magnitude and variety of transport emissions for the Danish market. The cases were selected for creating a variety in scale and typology, see Table 3 and Appendix A. The cases represent projects completed between 2015–2023. Their life cycle inventories were provided by consulting companies and were checked by Aalborg University. The cases also represent a variety in materiality and load-bearing frames to reflect deliveries from a variety of suppliers in different locations.

**Table 3.** Cases used for calculating emissions in module A4.

| Type Code | Building Typology | Quantity | GFA Range [m$^2$] |
|---|---|---|---|
| HC | Housing and commercial mix | - | - |
| AB | Apartment buildings | 1 | [2952] |
| HH | Detached homes and row houses | 2 | [179; 1954] |
| CO | Commercial buildings | 2 | [9630; 19,518] |
| EC | Education, care, culture | 2 | [860; 12,944] |
| OF | Offices | 2 | [1035; 6375] |
| | | Total 9 | |

### 2.3. Method for Module A5

In the present study, A5 includes carbon emissions associated with energy-consuming activities on site (both fuel and grid energy) as well as the carbon emissions associated with construction waste. The former is based on metered consumption data for heat, electricity and fuels. Since grid-based energy metering is a legal requirement, this data is seen as consistent and of high quality. Data was reported in different ways including the collection of monthly invoices or information directly taken from the supplier's customer system. Fuel consumption differs in several ways from grid-energy consumption. It is not linked to project expenses but rather to specific equipment, such as lifts, excavators or dumpers, with individual fuel tanks. Large projects have central diesel tanks for use by all contractors, both for larger or smaller machines like excavators or minor equipment (such as plate compactors or vibration rammers). Eventual fuel consumption associated with subcontractors has not been included in this study.

Regarding construction waste, this is structured into 16 waste categories that cover all main waste streams of the collected cases. It includes all on-site waste production such as packaging, damaged material, offcuts, surplus material and material from temporary structures. The waste stream quantities are based on legal waste-sorting requirements and are documented through invoices to the contractor or directly from the waste service company.

The quantification of emissions associated with the consumption of electricity, heating and fuel is based on the emission coefficients given in Table 4. Coefficients for grid energy are based on a national average in Denmark for 2020. Despite the wide adoption of electrical heat pumps in Denmark and a considerable share of the building stock still using natural gas, heat consumption in the reported data was district heating only. Available future policy scenarios used in current climate regulations [33] show that an expected future decrease of carbon emissions will reduce impacts from energy consumption considerably, i.e., by a factor 0.51 for electricity in 2025. The impacts for local district heat production in Denmark varies; however, verified data are not yet available for individual plants. The expected decrease in impacts for natural gas is related to a share of low-impact gasses, predominantly biogas at the moment. In the reported cases, all fuel consumption was diesel. The emission coefficient was calculated from the generic Ökobaudat database.

**Table 4.** Emission coefficients for impact assessment in module A5. Energy coefficients from 2020 are used in all cases. Scenarios are based on expected frozen policy and can be used to adjust this study's results in the future with correction factors.

| Resource | Unit | GWP Used in Study | | GWP Scenarios | | |
|---|---|---|---|---|---|---|
| | | 2020 [1] | 2023 [2] | Correction factor | 2025 [2] | Correction factor |
| Electricity | | 0.264 | 0.187 | 0.71 | 0.135 | 0.51 |
| District heating | kgCO$_2$e/kWh | 0.1314 | 0.105 | 0.80 | 0.0878 | 0.67 |
| Natural gas | | 0.2364 | 0.225 | 0.95 | 0.189 | 0.80 |
| Diesel emission | kgCO$_2$e/litre | 3.54 [3] | - | | - | |

[1] National Danish coefficient report [34]; [2] Updated scenarios from current Danish building regulations [33]; [3] Ökobaudat [35].

The quantification of impacts from construction waste are based on the estimated building products contained in the 16 defined waste categories (Table 5). This simplified classification was necessary, because no average data on the original product content of construction waste exists. The process was supported by contractors and waste handlers and a careful allocation of emission coefficients. Generic emission data was taken from the mandatory building regulations dataset [33]. This dataset is based on Ökobaudat, combined with branch-EPDs for wood and concrete. This database is used to calculate the A5 impacts related to wasted materials and products as the sum of impacts in modules A1–3 and C3–4. Transport of the initial fraction of materials and products to the construction site (before being wasted) in module A4 as well as the transport to waste handling in module A5 is not included, which underestimates the results in both modules insignificantly.

**Table 5.** Construction waste emissions ($kgCO_2e/kg$) based on the estimated share of contained building products and their generic GWP. Emission data are the sum of product and end-of-life stage (A1–3 + C3–4) from generic product datasets in the Ökobaudat database.

| Waste Category | GWP [$kgCO_2$/kg] | Share [%] | Selected Product Dataset |
|---|---|---|---|
| Metal | 1.96 | 90 | Steel profile |
| | | 10 | Aluminium profile |
| Mineral fibre | 1.58 | 100 | Mineral wool insulation |
| Combustible waste | 1.42 | 20 | Moisture membrane |
| | | 20 | OSB |
| | | 20 | Wood |
| | | 20 | Cardboard |
| | | 20 | EPS |
| Landfill | 0.84 | 80 | Concrete |
| | | 10 | Mineral wool insulation |
| | | 10 | PVC pipe |
| Unsorted construction waste | 0.48 | 56 | Concrete |
| | | 24 | Bricks |
| | | 10 | Mineral wool insulation |
| | | 2 | Moisture membrane |
| | | 2 | OSB |
| | | 2 | Wood |
| | | 2 | Roofing felt |
| | | 2 | EPS |
| Tile stone | 0.36 | 100 | Bricks |
| Gypsum | 0.33 | 100 | Gypsum fibre board |
| Wood | 0.25 | 10 | OSB |
| | | 90 | Wood |
| Stone materials | 0.23 | 70 | Concrete |
| | | 30 | Bricks |
| Concrete | 0.17 | 100 | Concrete |

Using generic data for the end-of-life stage in C3–4 is also a simplification and does not represent the actual treating procedure in the studied projects. This is not viewed as influencing the overall results significantly, because C3–4 normally is only a minor part of the product impacts, while production in A1–3 is dominating, in particular for waste categories with high quantities such as concrete and cementitious products. The advantage of this approach is that it resembles the method most likely to be used for regulation and limit values, because no data on actual waste treatment impacts are available at the project level.

The derived emission coefficients for energy and waste were then applied in a case study. An initial screening of 139 cases resulted in a final selection of 52 cases with a satisfying quality of measured resource and energy consumption for the module A5 study (Table 6). The case

location covers all Danish regions: Capital (28), Central Jutland (10), Southern Denmark (6), North Jutland (2), Sealand (1) and undisclosed (5). The buildings were completed between 2015–2023; their gross floor areas (GFA) range from 153 to 76,400 m$^2$, and they represent 6 different building uses. Data for 44 of these cases were collected from 10 contractors and 6 consulting companies. The remaining 8 cases originate from the evaluation campaign behind the voluntary sustainability class, a programme for testing and preparing LCA requirements in Denmark during 2020–2023 [32].

**Table 6.** Cases used for calculating emissions in module A5.

| Type Code | Building Typology | Quantity | GFA Range [m$^2$] |
|:---:|:---:|:---:|:---:|
| HC | Housing and commercial mix | 3 | [6440; 76,400] |
| AB | Apartment buildings | 8 | [528; 24,000] |
| HH | Detached homes and row houses | 22 | [153; 3266] |
| CO | Commercial buildings | 4 | [1035; 42,260] |
| EC | Education, care, culture | 9 | [210; 16,059] |
| OF | Offices | 6 | [1800; 15,120] |
| | | Total 52 | |

## 3. Results

### 3.1. Transport to Site (A4)

#### 3.1.1. Product Level

The results for the transport of materials are given both in terms of distance and GWP. The distances are specified in more detail for all 36 subgroups, reflecting the process of assessing the typical transport of the building materials for use on Danish construction sites (Figure 3). The distances vary depending on market availability and share between suppliers at different locations. Examples of products with dominating local production are concrete, reinforcement bars, some insulation types or bricks. Long-distance road transport often includes a share of import and can be found for natural stone, calcium-silicate or fibre cement. Shipped goods include gypsum boards, natural stone and PV modules.

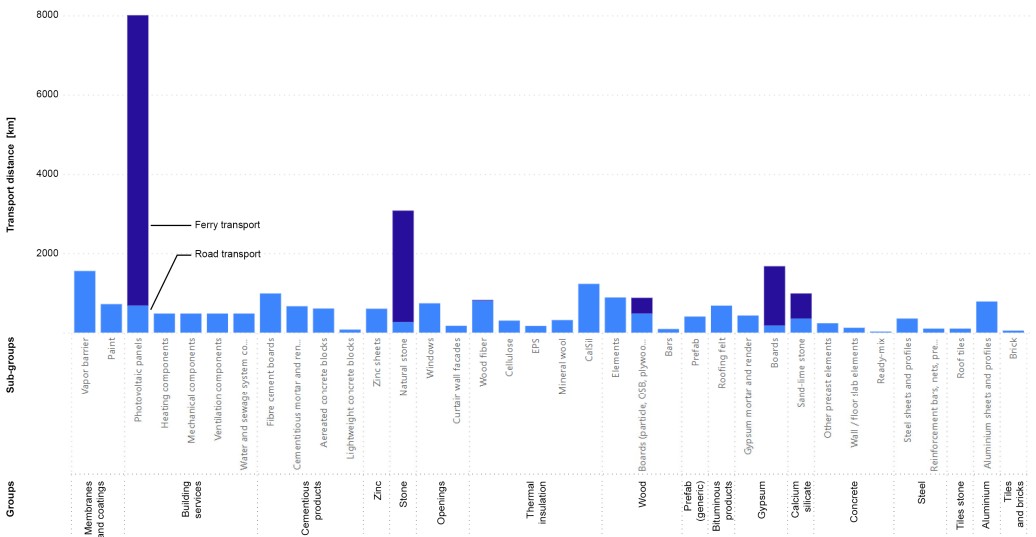

**Figure 3.** Average truck and ferry transport distances from factory to site at project subgroup level.

The GWP values for products calculated from the subgroup level are then averaged for the 17 product groups (Figure 4). The impacts vary between 0.00 and 0.41 kgCO$_2$e/kg.

The results at the product group level are based on a function of distance and transport form. Here, membranes and coatings as well as building services and natural stone have high impacts. The heaviest materials such as concrete, bricks and steel have low impacts due to local processing. This might seem misleading for metal products, since there is no raw material production in Denmark for this product type; however, the final processing step takes place in Denmark, which determines the location as the place of production in module A4.

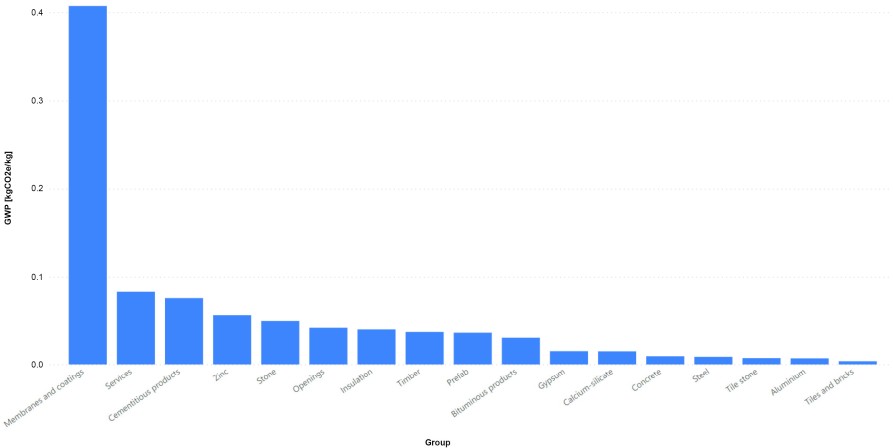

**Figure 4.** Transport emission coefficients for product groups including all occurring transport forms sorted by magnitude.

### 3.1.2. Building Level

The derived emission coefficients from the product group level are applied to the specific material quantities of nine construction cases (Figure 5). Emissions vary from 0.11 to 0.47 kgCO$_2$e/kgy. The most significant contributors are cementitious products, which include wide-spread aerated concrete blocks, due to their high weight values, used in many wall constructions. The project with the overall highest impacts is a mass timber building (200 CO). The main reason for this is the transport of cross-laminated timber (CLT) elements from Austria to Denmark, which is the standard supply chain for CLT.

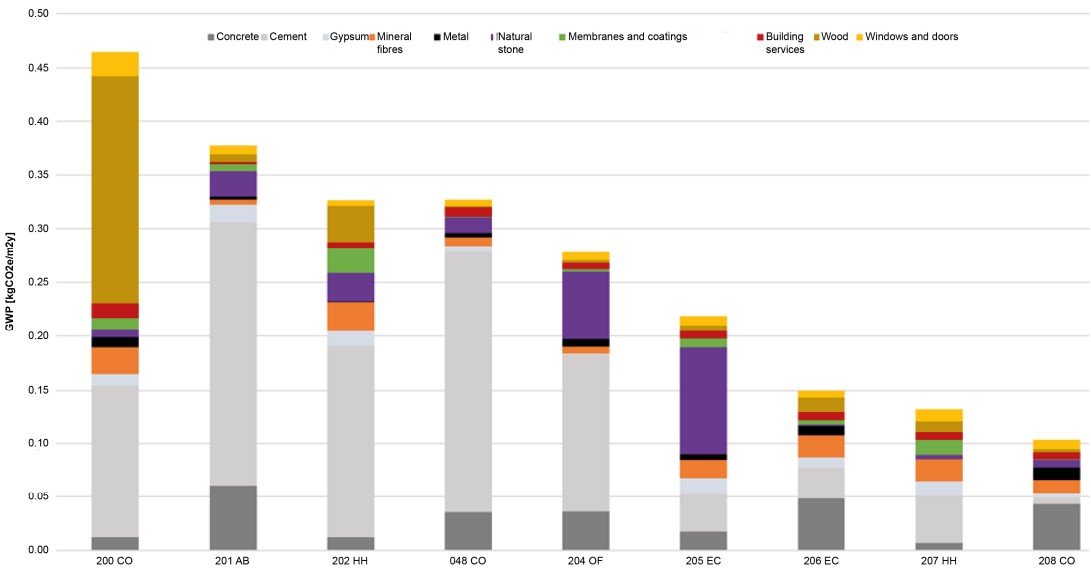

**Figure 5.** Climate impact for A4 transport, based on nine cases and divided into product groups used from the transport emission coefficients presented in the Section 2. Note: CO = Commercial building, AB = Apartment building, HH = Detached home and row house, EC = Education, care, culture, OF = Office.

Figure 6 shows the A4 impacts for the product groups and their variation. The variation within groups stem from the variation of mass in each project, where cementitious products vary most. This can be explained by the use of aerated concrete. It is used in large quantities in most low-rise buildings as well as wall material. In contrast, buildings using other wall materials have extremely small amounts of cementitious products.

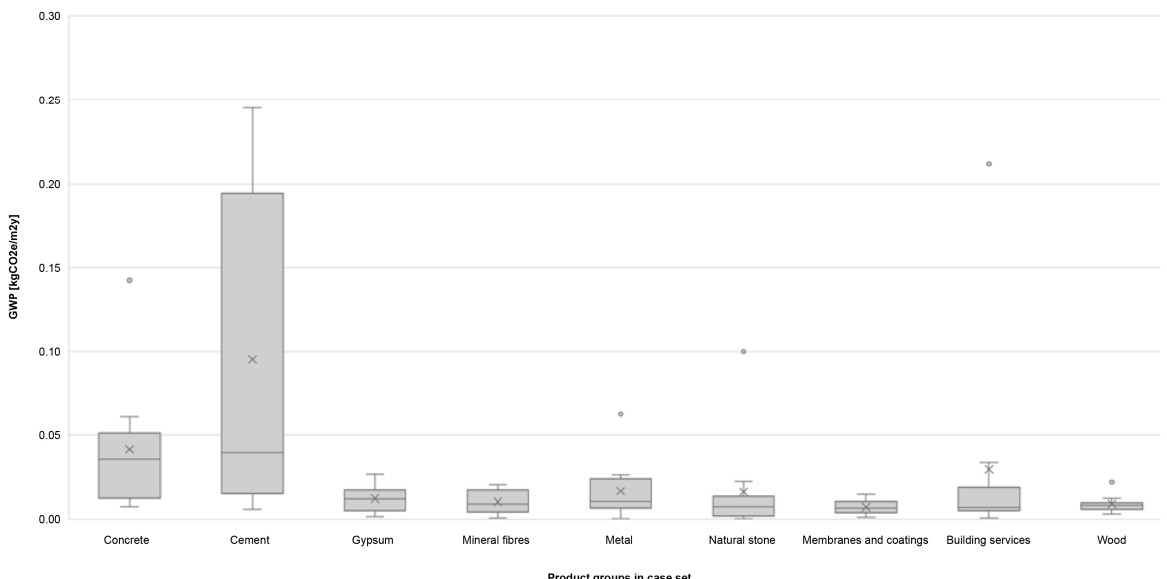

**Figure 6.** GWP boxplot for product groups for the 9 cases in the A4 set of cases. Calculations based on the transport emission coefficients.

### 3.2. Construction Installation Process (A5)

#### 3.2.1. Electricity

A total of 39 cases provided data on electricity consumption (Figure 7), ranging from 0.00 to 0.75 kgCO$_2$e/m$^2$y with a median at 0.19. The upper quartile contains ten detached homes or row houses, two commercial projects and one educational/case/culture project. The interquartiles consist of six detached homes or row houses, four apartment buildings, three offices, two apartment buildings, two housing and commercial mixes, two commercial projects and two educational/case/culture projects. The lower quartile contained nine detached homes or row houses and one apartment building.

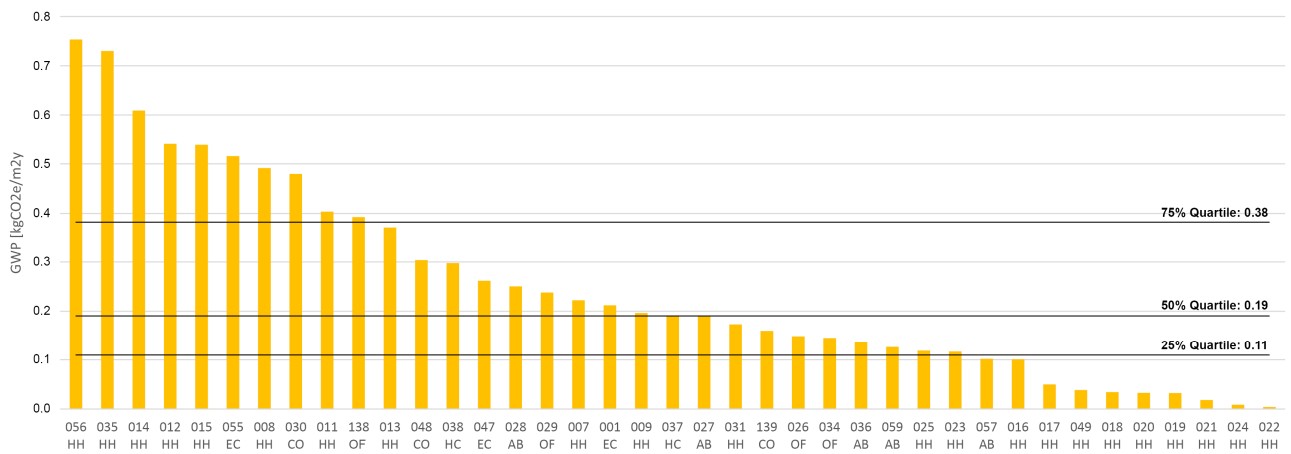

**Figure 7.** GWP of electricity consumption in the 39 cases with available data on electricity. Consumption also includes electricity-based heat sources. Note: CO = Commercial building, AB = Apartment building, HC = Housing and commercial mix, HH = Detached home and row house, EC = Education, care, culture, OF = Office.

### 3.2.2. Heating

Only 15 of the 52 cases provided information regarding heating due to five possible factors (Figure 8). In home buildings (type code HH), the district heating or natural gas suppliers often do not charge for consumption occurring during the construction process, thus before handover. Furthermore, the realisation of small projects with a construction period below one year may take place outside the heating season. The remaining factors may apply for all sorts of projects. Some projects use intermediate heating sources such as portable electrical room heaters during the construction stage; however, this practice is being phased out. Finally, heating is most often paid by the owner, which in most cases is not the contractor and, in consequence, may have been ignored by the contractor's data delivery. Therefore, blank data for heat consumption can certainly not be confirmed as being equal to no consumption. This is different to electricity, waste and fuel, which all have occurred with certainty, and blank data means a lack of reporting rather than no consumption.

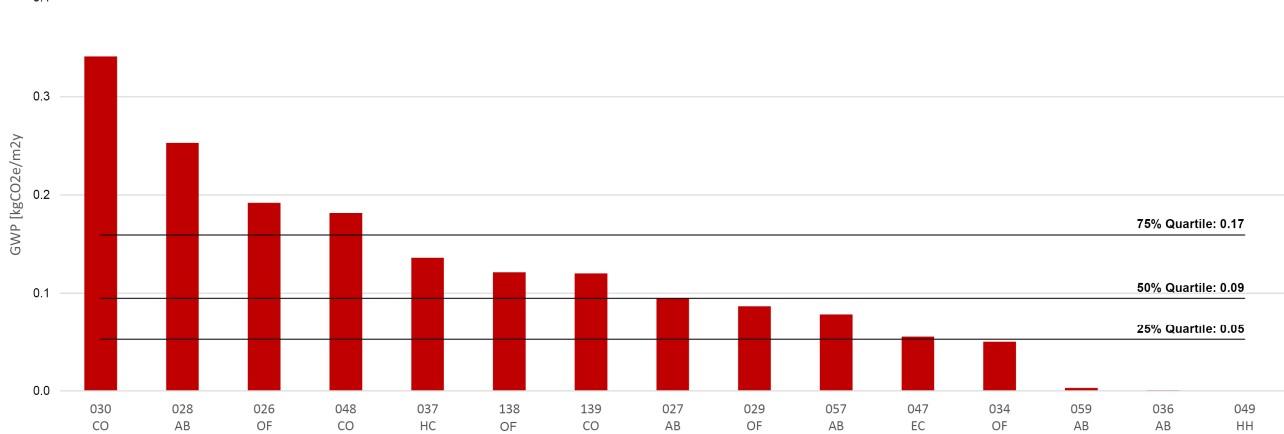

**Figure 8.** GWP of heat consumption in 15 cases with available data on heat (district heating). Consumption does not include electricity-based heat sources. Note: CO = Commercial building, AB = Apartment building, HC = Housing and commercial mix, HH = Detached home and row house, EC = Education, care, culture, OF = Office Fuel.

For the cases for which heating data were available, Figure 8 shows that emissions range from 0.00 to 0.34 $kgCO_2e/m^2y$ and have a 0.12 median value. The upper quartile contained two commercial projects, one apartment block and one office. The interquartiles contain two apartment blocks, one mixed-use project, one commercial project, one educational/care/culture project and one office. The lower quartile contains two offices and one detached home or row house project.

In 39 cases, data for fuel consumption was provided. Apart from a very small fraction of biofuel, all consumption originates from diesel. Emissions from on-site fuel consumption (Figure 9) range from 0.00 to 0.65 $kgCO_2e/m^2y$ with a 0.08 median value. In the upper quartile, we find five detached homes or row houses, two educational/care/culture projects, two commercial projects and one office. The interquartiles contain five detached houses or row house projects, three offices, two apartment blocks, two educational/care/culture projects and one mixed-use project. In the lower quartile, we can see five detached homes or row house projects, four educational/care/culture projects, one apartment block and one commercial project.

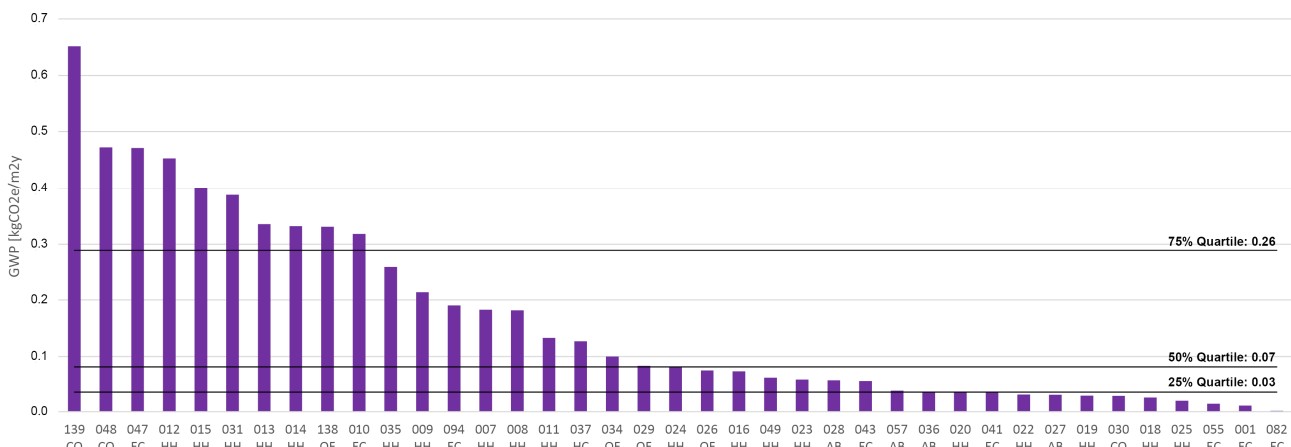

**Figure 9.** GWP of fuel consumption in 39 cases with available data on fuel. Fuel-based temporary heaters are included. Note: CO = Commercial building, AB = Apartment building, HC = Housing and commercial mix, HH = Detached home and row house, EC = Education, care, culture, OF = Office.

### 3.2.3. Waste

The GWP for construction waste was calculated as a function of waste quantity and the emission coefficients developed in Section 2. Along with the quantified results, this study also shows that the European waste catalogue for classifying waste in the European Union is not well integrated in the Danish construction industry. Moreover, the used terminology in the invoices varied not only between projects but also internally in each project and seems to depend on the individual truck driver. This inconsistency was also found by Lindhard et al. [36] who pointed out that the signage on the containers tends to vary even within project stages. This increases the degree of mis-sorted waste, while uncertain classification has complicated data analysis.

The waste data from 51 construction sites are presented in terms of waste category mass and emissions. A high variation of waste generation is registered when comparing rates across projects, in particular mixed-waste categories and combustible waste (Figure 10).

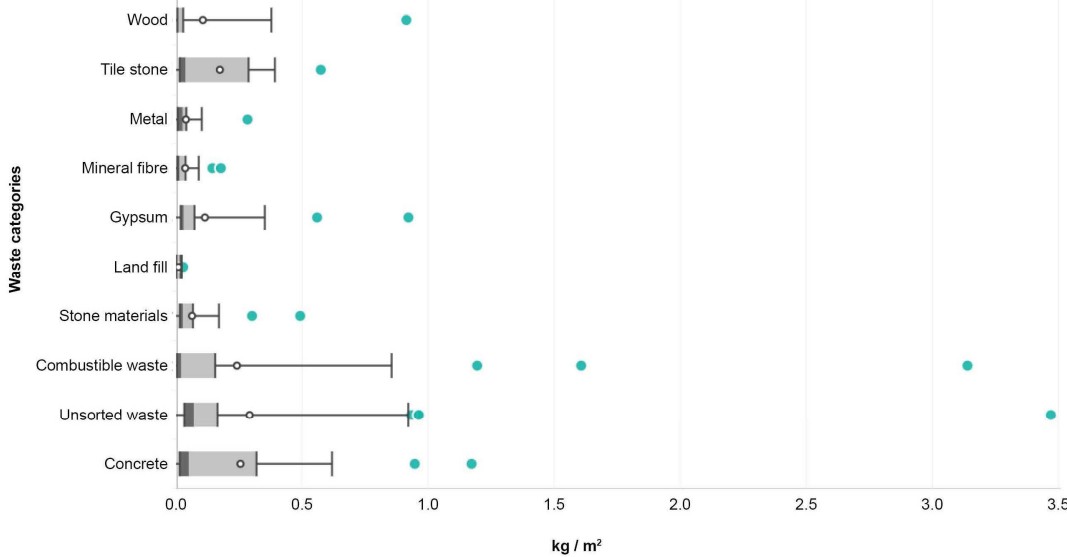

**Figure 10.** Boxplot diagram of waste fractions mass generated in the 51 cases with available data.

The emissions from waste generation (Figure 11) range between 0.12 and 1.85 kgCO$_2$e/m$^2$y with a 0.49 median value. The upper quartile contains four detached homes or row houses, two

educational/case/culture projects, two offices, one mixed project, one apartment block and one commercial project. The interquartiles contain 14 detached homes or row house projects, four educational/case/culture projects, two commercial projects, one mixed-use project and one apartment block. The lower quartile contains four apartment blocks, three detached homes or row houses, two offices, one mixed-use project and one educational/case/culture project.

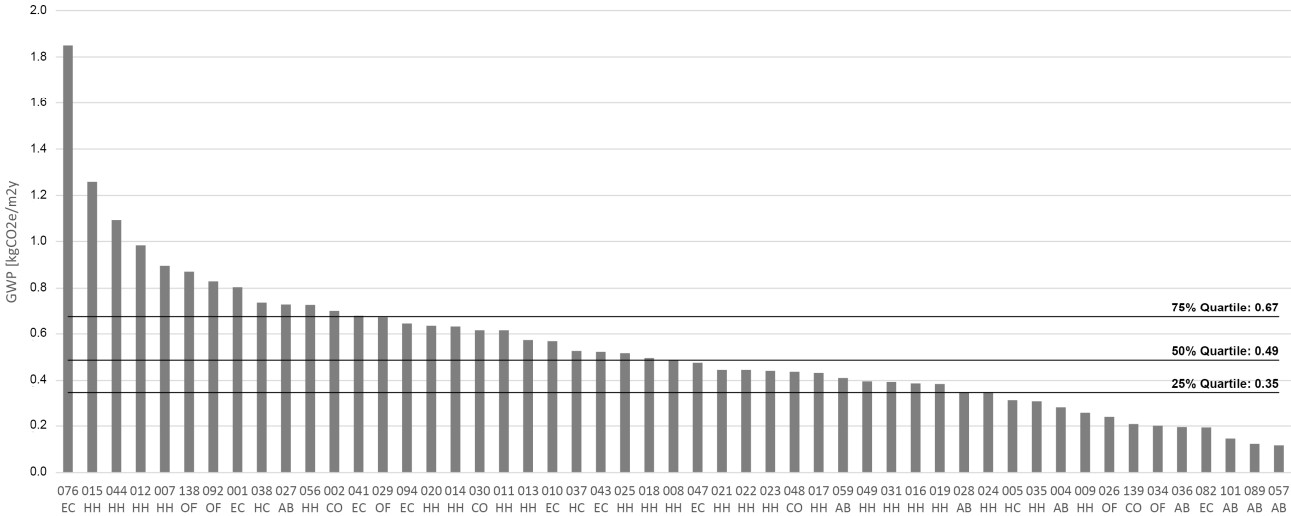

**Figure 11.** GWP of waste generation in the 51 cases with available data. Note: CO = Commercial building, AB = Apartment building, HC = Housing and commercial mix, HH = Detached home and row house, EC = Education, care, culture, OF = Office.

### 3.2.4. A5 in Total

The emissions from all cases with the reported consumption data are shown in Figure 12. The emissions vary between 0.12 and 2.20 kgCO$_2$e/m$^2$y with a median at 0.77. The overall median for A5 increases to 0.98 when the blank data for electricity, fuel and waste are corrected with the median data from cases with the reported data. To avoid an overestimation of emissions, the lacking heat data was left blank, since no consumption is a possibility.

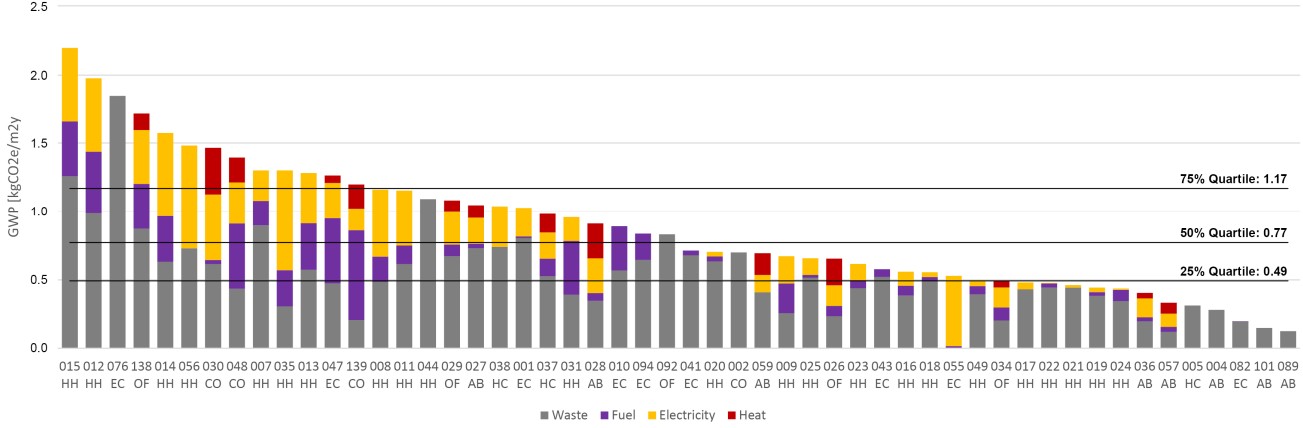

**Figure 12.** Total GWP of A5 for all cases. Lacking data is not adjusted but included with a value of 0. Case number and use code is given below the X-axis. Note: CO = Commercial building, AB = Apartment building, HC = Housing and commercial mix, HH = Detached home and row house, EC = Education, care, culture, OF = Office.

Even when corrected, the results show a considerable variation between the cases. A subsequent analysis (Figure 13) confirms that the variation is only poorly related with building use as such but is mostly caused by small projects. This is supposedly caused by

their short construction period and the over-proportional influence of seasonal energy demands depending on whether (or not) projects run during wintertime. The small variation across the remaining building uses is not necessarily related to use but many other known factors including soil conditions, basement volume, foundation methods or alike.

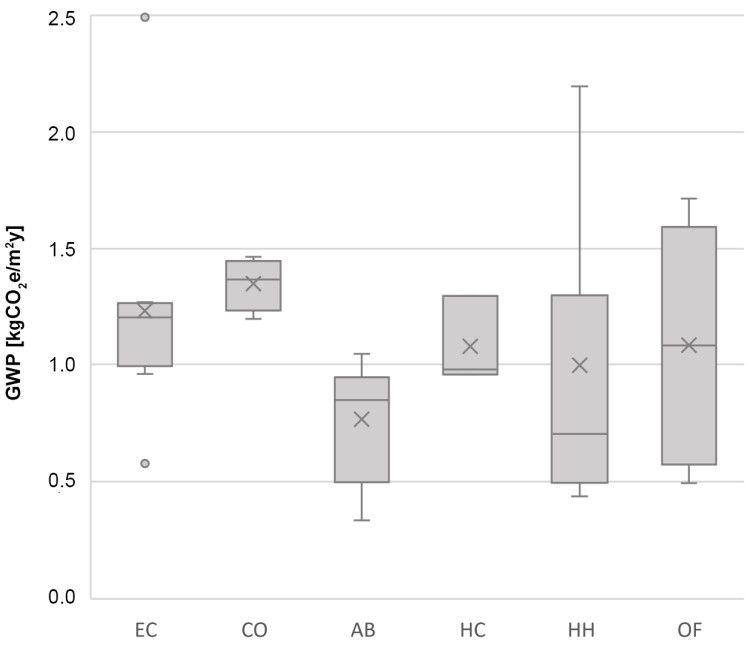

**Figure 13.** GWP boxplot as a function of building use. Results are based on the data as in Figure 12. Note: CO = Commercial building, AB = Apartment building, HC = Housing and commercial mix, HH = Detached home and row house, EC = Education, care, culture, OF = Office.

### 3.3. Total Results of A4 and A5

The key results for A4 and A5 are presented in Table 7. For each quartile, the results are first given individually for transport and the four resources; the latter are based only on the cases with reported data. The second column for each quartile represents the sum of each module. The third column shows the total sum of both modules in the construction process stage. The aggregated results are both given with and without the correction for missing data. Non-corrected data are shown secondarily in brackets because they are underestimating the total.

**Table 7.** Reference values for the climate impact for transport A4 and the construction installation process A5. All values are measured in $kgCO_2e/m^2y$. * Sum is adjusted for missing electricity, fuel and waste data.

|  |  | **25% Quartile** | **50% Quartile (Median)** | **75% Quartile** |
|---|---|---|---|---|
| Module A4 | Transport | 0.15 | 0.28 | 0.33 |
| Module A5 | Electricity | (0.11) | 0.19 | 0.38 |
|  | Heating | (0.07) | 0.12 | 0.18 |
|  | Fuel | (0.04) | 0.08 | 0.29 |
|  | Waste | (0.35) | 0.49 | 0.67 |
|  | Sum | 0.49 | 0.77 | 1.17 |
|  | Adjusted * | 0.56 | 0.96 | 1.21 |
| A4 + A5 | Sum | 0.64 | 1.05 | 1.50 |
|  | Adjusted * | 0.71 | 1.24 | 1.54 |

Data for A4 and A5, respectively, come from two sets of cases and rely on different methods. The A4 results include assumptions on average transport for the product groups to

Danish construction sites, which are then combined with real material inventories in nine cases. The emissions for A5 are all based on reported consumptions combined with assumptions on average product contents in the case of construction waste. Module A5 may slightly underestimate the true emissions in the studied cases due to the uncertainty of whether the lack of reported heat consumption in some cases equals to no actual heat consumption.

Heating has only been reported in 15 of the 52 cases; however, it is not clear where the missing data may indicate absent heat consumption or a lack of reporting. Also, construction projects do not necessarily require the final heat supply such as district heating, as it can be easily substituted with electrical- or fuel-based heating. Some construction periods may have taken place outside the heating season. We assume that at least parts of the data gap must be interpreted as a lack of reporting. This renders the presented emissions in Table 7 rather optimistic and the quartiles of A5 (0.56, 0.96 and 1.21) as representing a lower boundary of the true emissions.

To calculate an expected upper boundary of A5, the missing values of emissions due to heating have been replaced with median values. This creates a more conservative reference value because it is assumed that all cases with missing values used heating. As a result, the quartiles of A5 are increased to 0.66, 1.03 and 1.27. The lower and upper boundaries represent the outer limits of the expected emissions; thus, the true reference value is expected to lie between these. The difference in emissions between the upper and lower boundaries is 0.10, 0.07 and 0.06; thus, the difference is significant but low in relation to the overall A5 emission levels.

A qualified estimate to the true reference values for A5 would be the mean of the outer boundaries as this represents the middle value. Based on this approach, the quartiles are 0.61, 1.00 and 1.24.

## 4. Discussion

This study combines estimates with the measured data needed for A4 and A5 calculations to analyse a large sample of cases. In the A4 study, a main uncertainty lies in the development of emission coefficients for the product groups, both in estimating the average transport chains and conditions and in the selected product classification. The mapped transport forms and distances were then used to calculate emissions with the help of developed emission coefficients. These already include assumptions on the capacity utilization of transport vehicles but lack return journeys. In total, future research will analyse the influence and sensitivity of these factors for improving reliability. This will require more specific transport data from building suppliers. Another uncertainty is the distribution of materials and their correlated transport emissions in different building projects. The chosen sample of nine cases based on an assumed variation provides a first indication of emissions; however, more cases and in-depth analyses are necessary to assess the distribution of materials and equipment. This will also provide insight into potential strategies for mitigating transport emissions within the boundaries of construction projects. In terms of completeness, the transport of machinery to and from the site as well as the transport of waste to handling has been excluded.

As for the A5 module study, uncertainty lies in the impact assessment of construction waste, where the deviation between the developed coefficients and average or actual content of products is not known. Another approach used for assessing construction waste is subtracting the calculated material quantity during cost estimation or tendering from the purchased product quantity. In this study, no such data was collected; however, a subsequent study should compare both methods regarding accuracy and adequacy in terms of reporting and control from a regulation perspective. In terms of data completeness, some elements have been omitted in the calculation of the A5 module. The reporting of fuel consumption is assumedly incomplete in many cases with the available fuel data. This includes machinery not fuelled from the on-site tank. The use of on-site tanks is more common in large projects. This also includes fuel consumption from subcontractors. However, the main consumption is expected to be metered. The used source for diesel emissions has not been checked against

other sources. It can be assumed that other sources for well-to-wheels emissions are slightly lower than the ones used in the present study. Emissions from water were collected in 11 cases; however, the median GWP of 0.002 kgCO$_2$e/m$^2$y has shown its insignificance and it was omitted.

It can be concluded that the sum of the above-mentioned excluded aspects is estimated to have a negligible effect on the overall results in the A4 and A5 modules.

Using a reliable representation of the data's central tendency is important to ensure the validity of the findings. Based on the identified impacts related to the A4 and A5 modules, the calculated central tendency is presented in Table 8. The method applied to measure the central tendency is important as it can affect the findings. An example is the over-representation of detached homes and row houses in the A5 set of cases. Their emissions are mainly located in the upper and lower quartiles, especially for electricity. Thus, using a truncated or an interquartile mean will reduce the over-representation of this building use in the data.

**Table 8.** Measures of the central tendency of the data in the A5 module before replacing missing values with medians.

|  | Electricity | Heat | Fuel | Waste | Total |
|---|---|---|---|---|---|
| Mean | 0.25 | 0.13 | 0.16 | 0.54 | 0.88 |
| Median | 0.19 | 0.12 | 0.07 | 0.49 | 0.77 |
| Interquartile mean | 0.20 | 0.12 | 0.10 | 0.50 | 0.81 |
| Midrange | 0.38 | 0.17 | 0.33 | 0.98 | 1.20 |
| Truncated mean (−5 end values) | 0.22 | 0.12 | 0.13 | 0.51 | 0.84 |

Interestingly, there is a low variation between the different measurements of the central tendency. Thus, when looking at the emissions before replacing data gaps with medians, the different measurements of the central tendency only vary a little when disregarding the midrange. In all cases, the lowest value is the median while the highest value is the mean. Applying median values for replacing data gaps and as final reference values, might consequently underestimate the emissions. However, this effect is assessed to be minimal compared with the applying mean, interquartile or truncated mean values. The effect on total values when instead replaced with the mean is an increase in the central value from 1.03 to 1.04 kgCO$_2$e/m$^2$y.

The results align well with existing studies (Table 1), in particular, studies with multiple European cases [10,25]. The overall range of A4 and A5 emissions in previous studies and current standard values range between 0.05 and 0.72 and 0.45 and 1.43, respectively. In the present study, the median emission of A4 was estimated at 0.28, thus it is well within the expected range, which is in the lower end of the range but close to the middle. Our result is also 35% lower when compared to the A4 default value of 0.43 kgCO$_2$e/m$^2$y provided in Finland, which represents the value of 27 kgCO$_2$e/m$^2$ adjusted to a per year basis (considering a 50-year reference study period) and reduced by the applied top-up factor of +20% to all Finnish generic data [11]. The adjusted median emission level of A5 was estimated to be 1.00, again within the expected range laying in the middle of the range of the previously found emissions. Compared to the default values provided in Finland [11] and the UK [12]—the former country provides a range of 0.85–1.36 kgCO$_2$e/m$^2$y, depending on the building type (adjusted similarly to the A4 default value), and the latter country provides a range of 0.6–1.6 kgCO$_2$e/m$^2$y, depending on whether demolition of pre-construction occurs—our result again lies close to the middle towards the lower end of the range.

Two of the previous studies only included the summed emission of the A4 and A5 modules with a range from 0.80 to 0.87 [10,24]. The present findings indicate the total emissions to be higher with an emission level at 1.05 without considering the missing values and 1.28 (1.00 + 0.28) when adjusting for the missing values.

Some variation between emission levels is expected. The previous studies have been conducted in different countries with a different approach to construction, using different materials, infrastructure or methods. More international cases will improve this study and might allow for a statistical analysis of the key parameters influencing emissions. This would then allow for developing guidelines for mitigation. Yet, the overall findings are within the expected range and are considered valid and a strong indication to the general level of emissions in the A4 and A5 modules.

Considering that the median carbon emission value for buildings in Denmark is 9.5 $kgCO_2e/m^2y$ for a limited system boundary of A1–3, B4 and C3–4 [37], the addition of A4 and A5 increases this value by 13.47%. This demonstrates the significance of A4–5 in current construction practices. In terms of mitigation potential, the industry might achieve upfront reductions through local product manufacturing whenever available, reducing empty runs, using low carbon fuels and reducing construction waste.

A4 only makes up 3% of emissions; however, we should keep in mind that the mitigation potential for transport has a high beneficial effect as it belongs to the upfront emissions. Also, building life cycle emissions are expected to decrease due to regulations taking effect during the coming years, among others. This will reduce emissions from other life cycle stages, which in turn will increase the relative significance of A4 and A5. Finally, future studies and policies need to extend the scope of the included environmental impact categories and resource indicators for achieving a more realistic view of the construction stages and the building life cycle in general.

We believe that the presented method for developing reference values for construction processes can be adopted by most other countries. However, the resulted estimated emissions will vary due to a series of parameters and will require further context-specific studies. The key parameters for the installation process in module A5 include energy supply, topography, ground conditions, the climate, degree of prefabrication and mechanization. The reference values for transport in module A4 are also dependent on their context. Potentially influential parameters include the geographic distribution of the construction product supply, vehicle standards, road conditions and topography.

## 5. Conclusions

This study investigated 52 + 9 Danish construction sites regarding carbon emissions from transport in module A4 and the construction–installation process in module A5. The method for A4 included the development of average emissions for building product groups as a first step. This was achieved by combining data from the environmental product declarations, industry data and estimations where other data were unavailable. These A4 emission coefficients were then applied in nine cases with available bills of quantities. The median result of 0.28 $kgCO_2e/m^2y$, lying within the reported range of existing studies (0.05 and 0.72), makes up 3% of the Danish reference of 60 buildings, including the stages A1–3, B4, B6 and C3–4. Uncertainty was mainly expected in the estimation of the average transport distances and vehicle fuel consumption. The omitted aspects include return journeys and the transport of machines and waste.

For module A5, in total, 52 construction sites were investigated with a varying completeness of consumption reporting for electricity, fuels, heat and waste. The method for assessing grid-based energy consumption (electricity and district heating) in A5 was the combination of metered consumption with the national average emission coefficients. The fuel consumption was mostly based on central site tanks and was assessed using a generic diesel emission coefficient for diesel. The construction waste was calculated from the actual weight of the sorted categories from the cases and the developed emission factors for the waste categories based on assumptions on the included products. When adjusting for the missing values, the median value of the A5 emissions is 1.00 $kgCO_2e/m^2y$; this also lies within the results from existing studies (0.45 and 1.43). A5 makes up 11% of the reference and therefore is identified as the most significant in this study.

More research is necessary for increasing the studied database and for investigating gaps and uncertainties. However, this study provides the most comprehensive study of construction site emissions so far, and the overall results are evaluated to be a valid reference for future studies and regulative policy.

**Author Contributions:** Conceptualization, K.K.; methodology, J.M. and M.B.; validation, K.K. and S.M.L.; formal analysis, J.M.; investigation, K.K., J.M., S.M.L. and M.B.; resources, S.M.L.; data curation, K.K. and J.M.; writing—original draft, K.K., S.M.L. and M.B.; writing—review and editing, K.K., S.M.L. and M.B.; visualization, K.K. and J.M.; project administration, K.K. All authors have read and agreed to the published version of the manuscript.

**Funding:** This paper presents results from the project 'Ressourceforbrug på Byggepladsen' (Resource Consumption on Building Sites), which is being conducted 2021–2024 and funded by the Danish Authority of Social Services and Housing.

**Data Availability Statement:** Data sharing not applicable.

**Acknowledgments:** The authors wish to thank the companies participating in the project for sharing their data on resource consumption.

**Conflicts of Interest:** The authors declare no conflict of interest.

## Appendix A

**Table A1.** Case set for A4, sorted by ID.

| ID | GFA m$^2$ | Total A4 kgCO$_2$e/m$^2$y |
|---|---|---|
| 200 CO | 19,518 | 0.46 |
| 201 AB | 2592 | 0.38 |
| 202 HH | 179 | 0.33 |
| 048 CO | 1035 | 0.28 |
| 204 OF | 6375 | 0.22 |
| 205 EC | 12,944 | 0.19 |
| 206 EC | 860 | 0.15 |
| 207 HH | 1954 | 0.13 |
| 208 CO | 9630 | 0.10 |

## Appendix B

**Table A2.** Case set for A5, sorted by ID. Blank values mean no data is reported.

| ID | GFA m$^2$ | Electricity | Heat | Fuel | Waste | Total A5 |
|---|---|---|---|---|---|---|
| | | | | kgCO$_2$e/m$^2$y | | |
| 001 EC | 974 | 0.21 | | 0.01 | 0.80 | 1.03 |
| 002 CO | 42,260 | | | | 0.70 | 0.70 |
| 004 AB | 13,827 | | | | 0.28 | 0.28 |
| 005 HC | 44,000 | | | | 0.31 | 0.31 |
| 007 HH | 185 | 0.22 | | 0.18 | 0.90 | 1.30 |
| 008 HH | 200 | 0.49 | | 0.18 | 0.49 | 1.16 |
| 009 HH | 167 | 0.20 | | 0.21 | 0.26 | 0.67 |
| 010 EC | 210 | | | 0.32 | 0.57 | 0.89 |
| 011 HH | 345 | 0.40 | | 0.13 | 0.62 | 1.15 |
| 012 HH | 211 | 0.54 | | 0.45 | 0.98 | 1.98 |
| 013 HH | 220 | 0.37 | | 0.34 | 0.57 | 1.28 |
| 014 HH | 209 | 0.61 | | 0.33 | 0.63 | 1.57 |
| 015 HH | 185 | 0.54 | | 0.40 | 1.26 | 2.20 |

**Table A2.** *Cont.*

| ID | GFA | Electricity | Heat | Fuel | Waste | Total A5 |
|---|---|---|---|---|---|---|
| | m$^2$ | kgCO$_2$e/m$^2$y | | | | |
| 016 HH | 164 | 0.10 | | 0.07 | 0.39 | 0.56 |
| 017 HH | 159 | 0.05 | | | 0.43 | 0.48 |
| 018 HH | 172 | 0.03 | | 0.03 | 0.50 | 0.56 |
| 019 HH | 164 | 0.03 | | 0.03 | 0.38 | 0.44 |
| 020 HH | 175 | 0.03 | | 0.04 | 0.63 | 0.70 |
| 021 HH | 180 | 0.02 | | | 0.44 | 0.46 |
| 022 HH | 153 | 0.00 | | 0.03 | 0.44 | 0.48 |
| 023 HH | 170 | 0.12 | | 0.06 | 0.44 | 0.61 |
| 024 HH | 165 | 0.01 | | 0.08 | 0.35 | 0.44 |
| 025 HH | 167 | 0.12 | | 0.02 | 0.52 | 0.66 |
| 026 OF | 13,974 | 0.15 | 0.19 | 0.07 | 0.24 | 0.65 |
| 027 EB | 16,957 | 0.19 | 0.09 | 0.03 | 0.73 | 1.05 |
| 028 AB | 9195 | 0.25 | 0.25 | 0.06 | 0.35 | 0.91 |
| 029 OF | 5115 | 0.24 | 0.09 | 0.08 | 0.67 | 1.08 |
| 030 CO | 3176 | 0.48 | 0.34 | 0.03 | 0.62 | 1.46 |
| 031 HH | 3266 | 0.17 | | 0.39 | 0.39 | 0.95 |
| 034 OF | 15,120 | 0.15 | 0.05 | 0.10 | 0.20 | 0.49 |
| 035 HH | 7100 | 0.73 | | 0.26 | 0.31 | 1.30 |
| 036 AB | 24,000 | 0.14 | 0.04 | 0.04 | 0.20 | 0.41 |
| 037 HC | 76,400 | 0.19 | 0.14 | 0.13 | 0.53 | 0.98 |
| 038 HC | 6440 | 0.30 | | | 0.74 | 1.04 |
| 041 EC | 2271 | | | 0.03 | 0.68 | 0.71 |
| 043 EC | 1800 | | | 0.05 | 0.52 | 0.58 |
| 044 HH | 7450 | | | | 1.09 | 1.09 |
| 047 EC | 1350 | 0.26 | 0.06 | 0.47 | 0.48 | 1.26 |
| 048 CO | 1035 | 0.30 | 0.18 | 0.47 | 0.44 | 1.39 |
| 049 HH | 239 | 0.04 | 0.00 | 0.06 | 0.40 | 0.49 |
| 055 EC | 242 | 0.52 | | 0.01 | | 0.53 |
| 056 HH | 160 | 0.75 | | | 0.73 | 1.48 |
| 057 AB | 528 | 0.10 | 0.08 | 0.04 | 0.12 | 0.33 |
| 059 AB | 9174 | 0.13 | 0.16 | | 0.41 | 0.69 |
| 076 EC | 2344 | | | | 1.85 | 1.85 |
| 082 EC | 16,059 | | | 0.00 | 0.19 | 0.20 |
| 089 AB | 12,018 | | | | 0.12 | 0.12 |
| 092 OF | 1800 | | | | 0.83 | 0.83 |
| 094 EC | 1563 | | | 0.19 | 0.64 | 0.83 |
| 101 AB | 5047 | | | | 0.15 | 0.15 |
| 138 KB | 11,895 | 0.39 | 0.12 | 0.33 | 0.87 | 1.72 |
| 139 CO | 8600 | 0.16 | 0.18 | 0.65 | 0.21 | 1.20 |

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
