# Peer review of "Carbon Emissions during the Building Construction Phase: A Comprehensive Case Study of Construction Sites in Denmark"

_sustainability, doi:10.3390/su151410992_

Round 1

Reviewer 1 Report

This study presents an analysis of carbon emissions of 61 Danish construction sites based on their energy consumption, waste production and transport in modules A4 and A5. This has important guiding significance for the realization of green buildings. All of the following comments should be addressed in detail and responded to in an appropriate manner so I can recommend the paper for publication:

 1. The description in the introduction is incoherent and needs further modification.

2. Please adjust and optimize the format of the table.

3. The authors should explain or annotate the abbreviations at their first use site. It is beneficial to readers who are not familiar with this field. E.g. LCA, GFA?

4. Please improve the clarity and font size of Figure 1.

5. It is recommended to add a secondary heading. E.g. Overall approach and case selection, Method for module A4.

6. Please check the format of reference.

7. Please revise typos and grammatical errors in the full manuscript.

Great

Reviewer 2 Report

This paper attempts to analyze the pattern and magnitude of carbon emissions for a sample of Danish construction sites and gives practical implications. I have the following concerns that may help the authors further improve the paper.

First, in the literature review, although the authors did a relatively good job in comparing their study to previous similar studies, the review part lacks important recent studies such as those highlighting governance, social impacts, economic consequences, and financial constraints (e.g., Li et al., 2023) and those emphasizing smart city management (e.g., Kumar et al., 2022) involved in the process of green building construction.

Second, regarding methodology, the authors need to discuss the advantages and limitations of the adopted methods and possible alternative methods. The authors also need to strengthen the reason why choosing these construction sites as analytical laboratories to study the research questions under concern.

Third, turning to result interpretation, the findings of this paper should be presented and discussed in comparison to the extant literature, so that readers can instantly figure out the marginal contribution.

Fourth, in the discussion section, the authors should also add the limitation of this study and how their findings can be generalized to other regions and other scenarios.

Finally, the appendix should also include a brief description of the sample and the construction sites that belong to this sample. Subsample analysis is a must as the authors claim that they are better than previous studies due to a more comprehensive investigation of relevant issues.

Reference

Li, D., Liu, Y., Sun, M., Wang, X. and Xu, W. (2023), "Does venture-backed innovation support carbon neutrality?", China Finance Review International, Forthcoming. https://doi.org/10.1108/CFRI-12-2022-0253

Kumar, P., Sharma, R. and Bhaumik, S. (2022), “MCDA techniques used in optimization of weights and ratings of DRASTIC model for groundwater vulnerability assessment”, Data Science and Management, 5 (1), 28–41. https://doi.org/10.1016/j.dsm.2022.03.004

Can be improved by using more short and concise sentences.

Reviewer 3 Report

The manuscript titled “Carbon emissions during the building construction phase: a 2 comprehensive case study of construction sites in Denmark” is good research for sustainability in construction. However, the methodology is not clear and the number of cases is not enough.

Abstract

In line 15, the author should give the full name for the first time writing of each module such as A4, A5, B4, and B6.

Material and method

1) The author was not clearly explaining how come of the result or how to calculate the result of A4 and A5.

2) The result of A4 is analyzed from only nine cases and it is not covered for all Building typology grouped. It seems to be not enough for summarizing.

3) Table 4, please confirm the meaning of GWP that the author used in the manuscript. Should it be an emission factor (EF)?

According to US-EPA, the Global Warming Potential (GWP) is a measure of how much energy the emissions of 1 ton of a gas will absorb over a given period of time, relative to the emissions of 1 ton of carbon dioxide (CO2).

4) The gray boxes in Table 5 should give the meaning.

Result

1) Some data such as heat is not available in many cases, then the result in Figure 12 might be affected by data missing.

.Discussion

The literature on the emission in modules A4 and A5 is not well cited in the discussion. More references are needed and more comparison of the data to other similar studies.

Reviewer 4 Report

  1. What are the major contributors to carbon emissions in modules A4 and A5? Are there any specific measures that can be taken to reduce these emissions?
  2. How do floor area and building use affect carbon emissions in the construction industry? Are there other parameters that have a more significant influence?
  3. What are the potential policy implications of implementing modules A4 and A5 in whole-life carbon assessments? How can these findings contribute to environmental sustainability in the building sector?
  4. How were the generic emission coefficients developed, and how can they be applied in the building industry to increase feasibility?
  5. What are the considerations and challenges in using modules A4 and A5 in environmental product declarations?

Provide more detailed information on the methodology used for data collection, measurement techniques, and any assumptions made during the analysis. This will enhance the transparency and reproducibility of the study.

Elaborate on the significance of modules A4 and A5 in carbon assessments, explaining why they are important and how their inclusion can improve the accuracy of carbon assessments. This will strengthen the argument for their implementation in future assessments.

Expand the discussion on the usability of modules A4 and A5 in environmental product declarations, exploring the challenges and opportunities associated with their integration into existing frameworks.

Some sentences and expressions need to be polished, and unclear logic affects reading.

Round 2

Reviewer 1 Report

The authors have revised the manuscript.

Author Response

Thank you for your helpful comments and efforts

Reviewer 2 Report

Thank you for responding to my previous comments. I still spot some minor format issues in the revised version. E.g., the conclusion section is not in a seperate line. There are something wrong about the units for appendix tables. Please be more careful in preparing the final version.

Above average and fine, but need typo detection throughout the paper.

Author Response

Formatting and spelling have now been checked in more detail and revised throughout the paper, see tracked changes in the manuscript

Reviewer 3 Report

please check the column header and the data in Table 6, they are not in the line.

Author Response

Table 6 is now revised. Thank you for your helpful comments and efforts

Reviewer 4 Report

The modification is good, and the current version can be accepted after being approved by the editor.

Author Response

(The authors gave the same response as above.)
